# Think in Cloud, Look at Edges: Semantic-Driven Query Decomposition for Efficient Video Reasoning

**Wenhao Zou** [1 2 3] **Zhijie Cai** [1 4 5] **Minchen Yu** [1 5] **Zongshuai Zhang** [2] **Guangxu Zhu** [1 4 5 6]

## Abstract

Long video understanding faces a critical dilemma: cloud-based Large Multimodal Models (LMMs) offer superior reasoning but suffer from prohibitive bandwidth costs and latency, while edge-based solutions sacrifice perception accuracy for speed. Current collaborative approaches attempt to bridge this gap via similarity-based filtering, yet they treat complex queries as flat semantic vectors. We identify this as a fundamental flaw leading to "Semantic Submergence," where dominant visual features drown out subtle but logically critical cues. To solve this, we introduce SCOPE (Semantic Cloud-Orchestrated Perception at Edge). Shifting the paradigm to "Think in Cloud, Look at Edges," SCOPE utilizes a cloud LMM to decompose complex queries into a structured Directed Acyclic Graph (DAG). This "observation plan" guides the edge to retrieve evidence based on logical necessity rather than mere statistical similarity. Experiments on Video-MME and LongVideoBench demonstrate that SCOPE redefines the Pareto frontier, matching cloud-level accuracy with significantly lower transmission costs and outperforming state-of-the-art baselines on complex reasoning tasks.

## 1. Introduction

The integration of Large Language Models (LLMs) into Large Multimodal Models (LMMs) has revolutionized open-world perception and reasoning (Munasinghe et al., 2025; Cui et al., 2025; Achiam et al., 2023; Bai et al., 2025), making Long Video Understanding a pivotal task for embodied intelligence applications ranging from smart city surveillance to home robotics (Lu et al., 2025; Chen et al., 2025a; Diko et al., 2025; Han et al., 2025; Ma et al., 2025b). However, the massive spatio-temporal redundancy in long videos (Wang et al., 2024a; You et al., 2024) renders Pure-Cloud architectures—which rely on uploading high-definition streams to billion-parameter models—impractical in network-constrained scenarios due to prohibitive bandwidth costs and latency (Gündüz et al., 2022; Huang et al., 2025; Niu et al., 2025).

To mitigate transmission bottlenecks, Edge-Cloud Collaboration has emerged as a consensus solution (Liu et al., 2026; Chen et al., 2025b), typically adopting a "coarse-to-fine" strategy where edge devices select Top-K keyframes via lightweight models (e.g., CLIP) for cloud processing (Lu et al., 2025; Azad et al., 2025). However, we argue that this "Video-Centric" paradigm fundamentally fails by compressing video signals while neglecting the Query's guiding role; specifically, reliance on Global Semantic Similarity leads to a critical failure mode known as *Semantic Submergence* (Yuan et al., 2025).

Current flat "matching" strategies struggle with complex temporal dependencies (Shi et al., 2025; Ma et al., 2024; Jin et al., 2026). For instance, in the query "Did he turn off the light after cooking?", lightweight edge models tend to over-retrieve visually dominant concepts such as "cooking," while simultaneously suppressing subtler yet logically critical actions like "turning off the light" or transitional frames. We define this suppression as *Semantic Submergence*, where the edge's inability to infer compositional structure leads to a broken evidence chain, rendering even powerful cloud VLMs incapable of correct reasoning due to fractured context (Peng et al., 2017; Tang et al., 2025b; Liu et al., 2024; Yin et al., 2023).

To address this core conflict, we posit that complex reasoning requires "Planning" before "Compression." While linear Chain-of-Thought structures have shown promise (Wei et al., 2022), Directed Acyclic Graphs (DAGs) offer superior efficacy in modeling complex, non-sequential dependencies

[1]Shenzhen Research Institute of Big Data, Shenzhen, China [2]Institute of Computing Technology, Chinese Academy of Sciences, Beijing, China [3]University of Chinese Academy of Sciences, Beijing, China [4]Shenzhen International Center for Industrial and Applied Mathematics, Shenzhen, China [5]The Chinese University of Hong Kong, Shenzhen, China [6]Shenzhen Loop Area Institute, Shenzhen, China. Correspondence to: Guangxu Zhu <gxzhu@sribd.cn>, Minchen Yu <yuminchen@cuhk.edu.cn>, Zongshuai Zhang <zhangzongshuai@ict.ac.cn>.

*Proceedings of the 43rd International Conference on Machine Learning*, Seoul, South Korea. PMLR 306, 2026. Copyright 2026 by the author(s).

(Shao et al., 2025). Building on this insight, we propose the "Think Cloud, Look Edge" paradigm. In this framework, the cloud transforms from a passive receiver into an active Planner, decomposing complex queries into structured DAGs that serve as precise Multi-step Observation Plans. The edge then shifts from a blind compressor to a purposeful Executor; by independently retrieving evidence for specific atomic events such as the end of cooking or the action of turning off the light, we ensure the uploaded frame sequence maintains logical completeness. This differs from holistic similarity-based selection: the edge no longer scores frames against a single query vector, but retrieves evidence for multiple structured and importance-weighted sub-goals.

Based on this insight, we propose **SCOPE** (**S**emantic **C**loud-**O**rchestrated **P**erception at **E**dge), a semantic-driven edge-cloud collaborative reasoning framework. Our main contributions are summarized as follows: (1) **Framework Innovation:** We introduce a novel edge-cloud video understanding architecture that incorporates a Query Decomposition mechanism. By shifting the cloud's reasoning capability upstream to the sampling planning phase, we achieve precise reasoning for complex long videos with minimal bandwidth usage. (2) **Algorithm Design:** We design a Semantic-Driven DAG Generation and Budget Allocation Algorithm. This algorithm utilizes Chain-of-Thought (CoT) to construct logical graphs and dynamically allocates transmission budgets based on the information entropy of each sub-task, breaking the hard trade-off between bandwidth and accuracy found in traditional methods. (3) **Empirical Validation:** We conduct extensive evaluations on two challenging long-video benchmarks, Video-MME and LongVideoBench. Results show that our method significantly outperforms SOTA collaborative baselines in logical reasoning accuracy and achieves a superior trade-off between latency and performance.

## 2. Empirical Analysis and Motivations

### 2.1. The Latency-Accuracy Trade-off

To explore the performance boundaries of edge–cloud collaboration in realistic deployments, we benchmarked different architectures on the Video-MME dataset. We used an NVIDIA A100 GPU running Qwen2.5-VL-72B as the cloud node and a laptop equipped with a RTX 4080 Laptop GPU running Qwen2.5-VL-3B as the edge node. Videos were sampled at 1 FPS, and a unified TopK strategy was applied under transmission budgets of 16, 32, and 64 frames.

Figure 1 compares end-to-end latency and accuracy under three deployment paradigms using the same TopK-based keyframe selection at 1 FPS. End-to-end latency is measured as the sum of computation and communication time. Pure edge inference achieves the lowest latency, remaining below 26 s across all budgets, but its accuracy saturates below

*Table 1.* Accuracy comparison between structured (w/ Decomp.) and flat (w/o Decomp.) selection strategies under different frame budgets.

| Frame Budget | Accuracy (%) | |
| --- | --- | --- |
| | w/ Decomp. | w/o Decomp. |
| 16 frames | 66.04 | 62.93 |
| 32 frames | 68.41 | 65.74 |
| 64 frames | 70.03 | 68.56 |

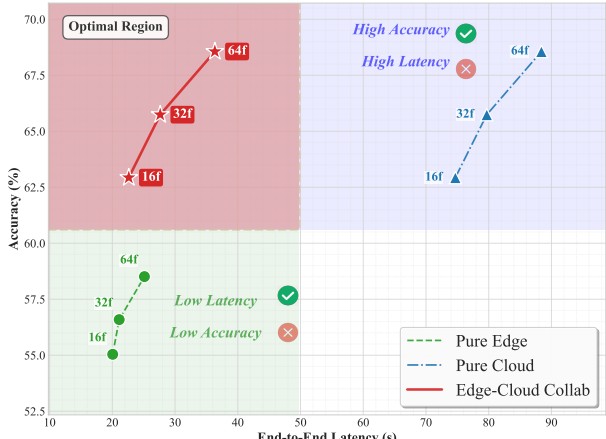

*Figure 1.* Latency–accuracy trade-offs under three deployment paradigms. All results use 1 FPS sampling and the same TopK-based keyframe selection under frame budgets of 16, 32, and 64.

60%. Pure cloud inference improves accuracy as the frame budget increases, reaching 68.56% at 64 frames, but incurs substantially higher latency, exceeding 70 s due to the cost of transmitting full video content. In contrast, the edge–cloud collaborative setting consistently matches cloud-level accuracy while reducing end-to-end latency by more than 50% relative to pure cloud inference, shifting the latency–accuracy frontier toward a more favorable operating region.

### 2.2. Analysis: Holistic vs. Structured Selection

To further investigate accuracy limitations, we compare **Holistic Top-K Selection** against **Structured Selection** under identical frame budgets. As shown in Table 1, holistic selection consistently lags behind the decomposed variant even with increased budgets, confirming that the bottleneck lies in imprecise guidance rather than insufficient frames. Consider the query asking what a man does after swimming; a holistic search often misses the distinct "refueling" action. This failure arises because the holistic embedding aggregates the primary event with various distractor concepts from the options, causing the subtle semantic signal for "refueling" to be diluted by the dominant features of "swimming" or other visually salient distractors. In contrast, SCOPE decomposes this into targeted sub-queries such as checking for the presence of vehicles or specific

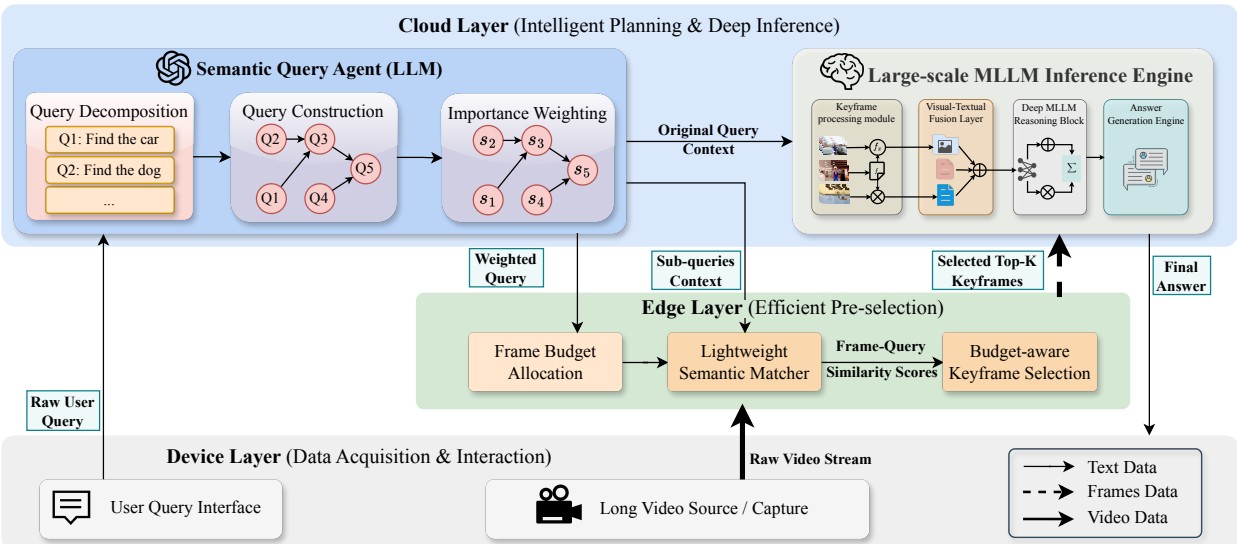

*Figure 2.* Overview of the SCOPE. The Cloud Agent plans the reasoning path via a DAG, while the Edge Node efficiently executes the plan through budget-aware frame selection.

post-swimming interactions. By retrieving and temporally organizing frames for these specific sub-goals, the system successfully captures the critical car-related evidence that leads to the correct answer. Detailed visualizations of this reasoning process are provided in **Appendix A**.

## 3. Method

### 3.1. Overview & System Architecture

To dismantle the severe trade-off between bandwidth and accuracy in long video understanding, we propose **SCOPE**, a semantic-driven edge-cloud collaborative reasoning framework. As illustrated in Figure 2, the system adheres to the paradigm of "*Think Cloud, Look Edge*" and consists of three core layers:

- **Cloud Layer (Intelligent Planning & Deep Inference):** Deploys large-scale multimodal large language models (MLLMs) and a Semantic Query Agent. It serves not only as the brain for final reasoning but also performs upstream "semantic planning" by decomposing the raw query into a weighted list of sub-queries.

- **Edge Layer (Efficient Pre-selection):** Deploys lightweight vision models (e.g., CLIP). Acting as the "executor," it receives planning instructions and performs budget apportionment, parallel semantic matching, and weighted keyframe selection.

- **Device Layer:** Responsible for video data acquisition and user interaction.

**Workflow Execution.** The inference process forms a closed loop: (1) The user inputs a query at the Device layer. (2) The Cloud Agent decomposes the query into a set of weighted sub-queries $\mathcal{P}$ and transmits them to the Edge. (3) The Edge first calculates the frame quota $\mathbf{b}$ for each sub-query using the Largest Remainder Method. (4) Subsequently, the Edge performs **parallel computation** of the similarity matrix between video frames and sub-queries, then sorts and truncates the results based on the quota $b_i$ to select keyframes. (5) Finally, the selected keyframe set $\mathcal{K}$ is uploaded to the Cloud, where the MLLM performs deep reasoning combined with the original query context to generate the final answer.

### 3.2. Cloud-Side Semantic Planning

Facing complex queries $q$, traditional holistic retrieval often leads to "semantic submergence." To address this, we introduce a Chain-of-Thought (CoT) based semantic planning module in the cloud.

**Query Decomposition.** We model decomposition as a generation task. Given a query $q$, the Cloud LLM generates a Directed Acyclic Graph (DAG) $\mathcal{G} = (\mathcal{V}, \mathcal{E})$, where each node $v_i \in \mathcal{V}$ represents an atomic sub-query, and edges $e_{ij} \in \mathcal{E}$ represent logical or temporal dependencies. Note that while the DAG structure helps the Cloud LLM clarify causal logic during generation, the resulting sub-queries are executed in parallel at the edge to maximize throughput.

**Semantic Importance Estimation.** Simultaneously, the LLM is instructed to estimate a semantic importance score $s_i \in \{1, \ldots, 10\}$ for each sub-query $v_i$. This discrete score reflects the necessity of the visual evidence corresponding to $v_i$ for answering $q$. The cloud ultimately issues the planning

instruction $\mathcal{P} = \{(v_i, s_i)\}_{i=1}^{N}$ to the edge.

### 3.2.1. INSTRUCTION DESIGN FOR QUERY DECOMPOSITION

To bridge the gap between abstract natural language queries and concrete visual evidence, we design a structured instruction mechanism that guides the Cloud LLM to function as a logical planner. The full system prompt used to instantiate this planner is provided in **Appendix C**. Here, we detail the three core logical constraints enforced during the decomposition process:

- **Decomposition:** The LLM is instructed to break down complex, multi-hop queries into *sub-queries*. Each sub-query must describe a single, visually verifiable event (e.g., splitting "Did he turn off the light after cooking?" into "Cooking activity" and "Turning off light action").

- **Logical Dependency via DAG:** To capture the temporal and causal relationships, the instruction requires the output to be structured as a Directed Acyclic Graph (DAG). An edge $v_i \rightarrow v_j$ signifies that event $v_i$ is a *logical precondition* or context for $v_j$. This structure preserves the narrative order of the video, enabling the reasoning engine to understand the sequence of events without relying on rigid timestamp filtering.

- **Importance Valuation:** A critical component of our prompt is the importance assessment. The LLM must assign a score $s_i \in \{1, \dots, 10\}$ to each node based on its contribution to the final answer. This transforms the query from a flat text string into a *weighted evidence map*, directly guiding the edge-side budget allocation.

By enforcing these logical constraints, the planner ensures that the subsequent retrieval process is both semantically comprehensive and resource-aware.

## 3.3. Edge-Side Budget Apportionment & Execution

The core task of the edge node is to maximize model utility under limited resources based on the instruction $\mathcal{P}$.

### 3.3.1. PROBLEM FORMULATION

We formulate the system as an accuracy maximization problem subject to latency constraints. Let $\text{Acc}(q, V; \mathcal{K})$ denote the answer accuracy given video $V$ and the uploaded keyframe set $\mathcal{K}$. The end-to-end latency $\text{Latency}(\mathcal{K})$ can be decomposed as:

$$\text{Latency}(\mathcal{K}) = T_{\text{plan}} + T_{\text{edge}} + T_{\text{tx}}(\mathcal{K}) + T_{\text{infer}}(\mathcal{K}) \quad (1)$$

where $T_{\text{plan}}$ is the cloud planning time and $T_{\text{edge}}$ is the edge processing time. The critical bottlenecks are the transmission latency $T_{\text{tx}}$ and cloud inference latency $T_{\text{infer}}$, both of

which are positively correlated with the total number of uploaded frames $|\mathcal{K}|$. Thus, satisfying the latency upper bound $D_{max}$ is equivalent to constraining the total frame count within a budget $B$.

Given the total budget $B$, we need to determine the frame allocation $b_i$ for each sub-query $v_i$ and the specific frame set $\mathcal{K}_i$ to maximize the comprehensive utility:

$$\max_{\{\mathbf{b}, \{\mathcal{K}_i\}\}} \quad \text{Acc}\left(q, V; \bigcup_{i=1}^{N} \mathcal{K}_i\right) \quad (2)$$

$$\text{s.t.} \quad \sum_{i=1}^{N} b_i \leq B,$$
$$b_i \in \mathbb{Z}_{\geq 0}, \quad b_i \geq b_{\min}, \quad \forall i,$$
$$|\mathcal{K}_i| = b_i, \quad \mathcal{K}_i \subseteq \mathcal{F}$$

where $b_{\min}$ is the minimum guaranteed frame count to prevent sub-query starvation.

### 3.3.2. PROBLEM TRANSFORMATION

Solving the above problem directly is prohibitively challenging because the optimization objective $\text{Acc}(\cdot)$ entails a complex deep model inference process without an analytical expression and is non-differentiable; furthermore, the combinatorial space of $\mathcal{K}_i$ grows exponentially.

To tackle this, we **recast** the optimization problem as an **Integer Seat Apportionment Problem**. Our core strategy is to maximize semantic coverage efficiency: maximizing the approximation of the actual allocated frames $b_i$ to the ideal quota $\tilde{b}_i$ derived from semantic importance $s_i$, subject to integer constraints. The objective transforms into minimizing the allocation deviation:

$$\min_{\mathbf{b}} \sum_{i=1}^{N} \left(b_i - \tilde{b}_i\right)^2 \quad \text{s.t.} \quad \sum b_i = B, b_i \in \mathbb{Z} \quad (3)$$

**Rationale:** We posit that the importance score $s_i$ serves as a proxy for the information entropy of the $i$-th sub-query. Allocating frames proportionally to $s_i$ effectively maximizes the total expected information gain captured by the edge node, thereby theoretically maximizing the upper bound of the cloud model's reasoning accuracy.

### 3.3.3. ALGORITHM: SEMANTIC-DRIVEN BUDGET ALLOCATION & RETRIEVAL

To efficiently solve this problem and execute retrieval on the edge, we propose a two-phase algorithm comprising **Budgeting** and **Parallel Retrieval**.

**Phase 1: Deterministic Budget Allocation.** As detailed in **Algorithm 1**, we employ the **Largest Remainder**

**Algorithm 1** Semantic-Driven Frame Budget Allocation

---

**Input:** Total Budget $B$; Min Guarantee $b_{\min}$; Importance scores $\mathcal{S} = \{s_1, \ldots, s_N\}$
**Output:** Integer allocation vector $\mathbf{b} = \{b_1, \ldots, b_N\}$

1: **Step 1: Initialization & Base Allocation**
2: Calculate normalized weights: $w_i \leftarrow s_i / \sum_j s_j$
3: Calculate remaining budget: $B_{\text{rem}} \leftarrow B - N \cdot b_{\min}$
4: **if** $B_{\text{rem}} < 0$ **then**
5:     **return** UNIFORMALLOCATION($B$)
6: **end if**
7: **Step 2: Ideal Quota Calculation**
8: **for** $i \leftarrow 1$ **to** $N$ **do**
9:     Ideal extra quota: $\tilde{e}_i \leftarrow B_{\text{rem}} \cdot w_i$
10:    Integer part: $e_i \leftarrow \lfloor \tilde{e}_i \rfloor$
11:    Fractional part: $r_i \leftarrow \tilde{e}_i - e_i$
12: **end for**
13: **Step 3: Largest Remainder Distribution**
14: Calculate unallocated remainder: $R \leftarrow B_{\text{rem}} - \sum_i e_i$
15: Get indices sorted by $r_i$ descending: $idx\_sorted \leftarrow$ ARGSORTDESC($\{r_1, \ldots, r_N\}$)
16: Initialize $\mathbf{b}$ with base guarantee: $b_i \leftarrow b_{\min} + e_i$
17: **for** $k \leftarrow 1$ **to** $R$ **do**
18:    $target\_idx \leftarrow idx\_sorted[k]$
19:    $b_{target\_idx} \leftarrow b_{target\_idx} + 1$
20: **end for**
21: **return** $\mathbf{b}$

---

**Method** (a.k.a. Hamilton Method) to determine the integer frame count $b_i$. Specifically, we first calculate the normalized weight $w_i = s_i / \sum_j s_j$. After deducting the base guarantee $N \cdot b_{\min}$, we calculate the remaining budget $B_{\text{rem}}$ and the "ideal extra quota" $\tilde{e}_i = B_{\text{rem}} \cdot w_i$ for each sub-query. To address the integer constraint, we decompose $\tilde{e}_i$ into an integer part $e_i = \lfloor \tilde{e}_i \rfloor$ and a fractional part $r_i = \tilde{e}_i - e_i$. The algorithm first allocates the undisputed integer part $e_i$. For the remaining unallocated budget $R = B_{\text{rem}} - \sum e_i$, we **sort the sub-queries by their fractional parts** $r_i$ in descending order and allocate one additional frame to the top-$R$ sub-queries. This yields the final allocation vector $\mathbf{b} = \{b_1, \ldots, b_N\}$.

**Phase 2: Parallel Semantic Matching & Selection.** With the budget $\mathbf{b}$ determined, the edge device executes the actual frame retrieval. To minimize $T_{\text{edge}}$, we leverage the matrix operation capabilities of modern deep learning frameworks to implement **system-level batch optimization**. We encode the $N$ textual sub-queries into a matrix $Q_{emb} \in \mathbb{R}^{N \times D}$ and the video frame stream into $V_{emb} \in \mathbb{R}^{T \times D}$. A single matrix multiplication $S = V_{emb} \cdot Q_{emb}^T$ yields the similarity score matrix for all sub-queries in parallel. To avoid redundancy among high-scoring adjacent frames and ensure visual diversity, we employ **Maximal Marginal Relevance (MMR)** for the final selection. For the $i$-th sub-query, we iteratively select $b_i$ frames that maximize the trade-off between relevance (score in $S$) and diversity. This design also mitigates semantically overlapping sub-queries: the planner is constrained to produce atomic visual checks, while MMR suppresses repeated evidence during edge-side retrieval.

**Fusion & Upload:** Selected frame indices are aggregated into a set $\mathcal{K} = \bigcup \mathcal{K}_i$. We **deduplicate** and **temporally sort** $\mathcal{K}$ to reconstruct the narrative before uploading.

**Complexity Analysis:** We analyze the computational overhead of SCOPE phase-by-phase. The budget allocation (Phase 1) involves sorting the fractional parts, with a complexity of $\mathcal{O}(N \log N)$. Since the number of sub-queries $N$ is typically small (e.g., $N \leq 10$), this step is negligible. The bottleneck lies in the similarity computation (Phase 2), which has a complexity of $\mathcal{O}(T \cdot N \cdot D)$. However, thanks to the parallel matrix multiplication on GPUs, the inference time remains constant with respect to the batch size $N$. Crucially, the planning time $T_{\text{plan}}$ in the cloud involves only text generation, which takes mere milliseconds and is negligible compared to the video transmission time $T_{\text{tx}}$.

## 4. Experimental Setups

**Real-world Heterogeneous Testbed.** All experiments were conducted in a real-world heterogeneous computing environment to ensure practical relevance. Our testbed consists of three physical nodes: a **Client Node** (standard laptop) responsible for video acquisition, connected via WiFi 6 LAN to the Edge; an **Edge Node** (high-performance laptop with NVIDIA RTX 4080 Laptop GPU) executing lightweight vision models and budget allocation; and a **Cloud Node** (server with $4\times$ NVIDIA A100 GPUs) hosting large-scale model inference. In our setup, the effective uplink throughput of the WiFi 6 LAN, measured end-to-end during experiments, ranges from 600 to 800 Mbps, while the uplink bandwidth between the Edge and Cloud nodes over the Wide Area Network (WAN) is measured to be 16–24 Mbps. These bandwidth regimes are consistent with commonly reported effective throughput ranges in real-world WiFi 6 and commercial cellular or broadband uplink deployments (Liu & Choi, 2023; Ghoshal et al., 2025).

**Models and Benchmarks.** For model configuration, we employed lightweight ViT architectures, specifically CLIP and BLIP, for visual preprocessing at the edge. The Qwen3-next-80b-a3b-instruct model serves as the semantic planner. For the video question answering (VQA) inference phase, we evaluated the **Qwen2.5-VL** (Bai et al., 2025) series across multiple scales, including 3B, 7B, 32B, and 72B-Instruct versions. We selected **Video-MME** (Fu et al., 2025) and **LongVideoBench** (Wu et al., 2024) as our evaluation benchmarks, as they provide comprehensive

*Table 2.* **Video-MME Results.** Performance across Small, Medium, and Large video durations. **Bold** indicates the best performance, and underline indicates the second best.

| Model | Method | K = 16 | | | | K = 32 | | | | K = 64 | | | |
|---|---|---|---|---|---|---|---|---|---|---|---|---|---|
| | | Small | Medium | Large | **Avg.** | Small | Medium | Large | **Avg.** | Small | Medium | Large | **Avg.** |
| **3B** | UNI | 65.2 | 52.2 | 46.0 | 54.5 | 66.6 | 55.6 | 48.0 | 56.7 | **70.0** | 57.4 | 50.0 | 59.1 |
| | TOPK | 64.1 | 52.4 | 48.6 | 55.0 | 67.4 | 54.7 | 47.7 | 56.6 | 69.1 | 56.2 | 50.2 | 58.5 |
| | AKS | 64.0 | 53.8 | 48.1 | 55.3 | 67.0 | 56.2 | 49.3 | 57.5 | 69.7 | 57.7 | 50.2 | 59.2 |
| | **SCOPE** | **66.1** | **54.1** | **48.7** | **56.3** | **68.2** | **58.1** | **51.7** | **59.3** | 69.9 | **58.2** | **52.1** | **60.1** |
| **7B** | UNI | 66.1 | 54.1 | 48.0 | 56.1 | 71.1 | 59.3 | 51.8 | 60.7 | **74.4** | 60.0 | 53.3 | 62.6 |
| | TOPK | 66.9 | 53.6 | 47.3 | 55.9 | 71.6 | 57.2 | 49.6 | 59.4 | 73.2 | 59.6 | 51.0 | 61.3 |
| | AKS | 69.3 | 57.1 | 49.2 | 58.6 | 71.2 | 60.4 | 51.2 | 61.0 | 73.0 | 60.9 | 53.6 | 62.5 |
| | **SCOPE** | **70.3** | **58.9** | **52.1** | **60.4** | **72.2** | 59.8 | **53.6** | **61.9** | 74.0 | **62.7** | **54.6** | **63.7** |
| **32B** | UNI | 70.8 | 57.7 | 51.4 | 60.0 | 72.1 | 61.7 | 54.2 | 62.7 | 75.3 | 65.4 | 55.8 | 65.5 |
| | TOPK | 68.8 | 57.0 | 52.2 | 59.3 | 73.4 | 59.7 | 51.8 | 61.6 | **76.4** | 63.6 | 54.4 | 64.8 |
| | AKS | 71.2 | 58.3 | 52.0 | 60.5 | 73.9 | 63.6 | 55.2 | 64.2 | 75.0 | 65.9 | 57.8 | 66.2 |
| | **SCOPE** | **72.1** | **61.4** | **55.6** | **63.0** | **75.4** | 63.1 | **57.0** | **65.2** | 76.1 | **67.1** | **58.2** | **67.1** |
| **72B** | UNI | 70.8 | 62.2 | 54.8 | 62.6 | 74.7 | 65.7 | 57.1 | 65.8 | 78.4 | 69.8 | 60.1 | 69.4 |
| | TOPK | 70.7 | 63.0 | 55.1 | 62.9 | 75.4 | 64.3 | 57.4 | 65.7 | 78.1 | 68.6 | 59.0 | 68.6 |
| | AKS | 72.8 | 63.9 | 56.0 | 64.2 | 75.4 | 66.7 | 59.2 | 67.1 | 78.2 | 70.3 | 59.6 | 69.4 |
| | **SCOPE** | **74.1** | **65.0** | **59.0** | **66.0** | **76.1** | **68.3** | **60.8** | **68.4** | **79.3** | 69.7 | **61.1** | **70.0** |

coverage of long-duration videos requiring complex reasoning. Under the default 1 FPS sampling rate, Video-MME contains 1021/489/2680 candidate frames on average/median/90th percentile, while LongVideoBench contains 477/210/1261 candidate frames. The dominant video resolution in both datasets is 1280×720, with a small number of lower-resolution videos; selected keyframes are transmitted at their original resolution unless otherwise specified. All reported results are averaged over the corresponding benchmark samples rather than measured on a single video.

**Baselines.** We benchmark SCOPE against three representative strategies: (1) **Uniform**, which samples frames at equal intervals, serving as a context-agnostic baseline; (2) **Top-K**, which selects frames based on cosine similarity scores calculated by the edge ViT model, representing standard semantic matching approaches; and (3) **AKS** (Tang et al., 2025a), a state-of-the-art keyframe selection method in long video understanding.

**Implementation Details.** Videos were sampled at a default rate of 1 FPS; experimental results under other frame rates exhibited similar trends and are detailed in the Appendix. Regarding inference optimization, models were loaded with automatic device mapping and Flash Attention 2 acceleration. To ensure reproducibility, we enforced deterministic generation by setting the temperature to 0, disabling random sampling, utilizing a beam size of 1, and setting the maximum generation length to 4096 tokens.

## 5. Experimental Results

### 5.1. Main Results

We comprehensively evaluated SCOPE on two benchmarks, Video-MME and LongVideoBench, across varying model scales and transmission budgets. Tables 2 and 3 present the quantitative comparisons against baselines.

**Results on Video-MME.** We first evaluate the performance on Video-MME across varying model scales and transmission budgets. As detailed in Table 2, SCOPE achieves consistent performance leadership across almost all experimental settings. The advantage is particularly evident under strict bandwidth constraints. For instance, with the 72B model at a budget of $K = 16$, SCOPE surpasses the SOTA method (AKS) by 1.8% and outperforms the Uniform baseline by 3.4%; with the 32B model under the same budget, SCOPE improves over AKS by 2.5%. This trend holds true from the lightweight 3B model to the large-scale 72B model, validating that our cloud-based planning effectively identifies high-value "golden frames" regardless of the visual backbone's capacity. To further investigate the source of these gains, we analyzed the fine-grained competency distribution using the representative setting of the 72B model at $K = 32$. As illustrated in Figure 3, SCOPE envelopes baseline methods across all six dimensions. It shows particular dominance in logic-intensive domains such as *Film & Television* and *Knowledge*, confirming that the Chain-of-Thought planning mechanism is crucial for disentangling complex narrative dependencies.

*Table 3.* **LongVideoBench Results.** Fine-grained duration analysis (8-15s to 900-3600s). **Bold** indicates the best performance, and underline indicates the second best.

| Model | Method | K = 16 | | | | | K = 32 | | | | | K = 64 | | | | |
|---|---|---|---|---|---|---|---|---|---|---|---|---|---|---|---|---|
| | | 8-15 | 15-60 | 180-600 | 900+ | Avg. | 8-15 | 15-60 | 180-600 | 900+ | Avg. | 8-15 | 15-60 | 180-600 | 900+ | Avg. |
| **3B** | UNI | **66.1** | 68.6 | 49.5 | 44.7 | 52.3 | **68.8** | 71.5 | 53.6 | 46.1 | 54.9 | **68.8** | **74.4** | 56.6 | 45.4 | 55.9 |
| | TOPK | 65.6 | **70.3** | 57.3 | 49.6 | 56.9 | 65.6 | 70.9 | 58.5 | 49.5 | 57.3 | 65.6 | 70.3 | 57.3 | 48.4 | 56.4 |
| | AKS | 65.6 | 67.4 | 56.6 | 48.9 | 56.0 | 65.6 | 69.2 | 56.3 | 49.3 | 56.3 | 65.6 | 70.3 | 55.8 | **51.1** | 57.1 |
| | **SCOPE** | 65.1 | **70.3** | **57.8** | **50.4** | **57.3** | 66.1 | **72.1** | **58.7** | **51.8** | **58.6** | 65.6 | 70.9 | **58.7** | 50.0 | **57.6** |
| **7B** | UNI | **73.5** | 68.0 | 54.6 | 47.9 | 56.2 | **73.5** | **75.0** | 57.5 | 50.0 | 58.9 | **75.7** | **80.2** | 58.0 | 49.5 | 59.8 |
| | TOPK | 65.6 | 69.8 | 61.4 | **53.7** | 59.8 | 65.6 | 70.9 | **63.1** | 55.0 | **61.0** | 65.6 | 73.3 | **62.9** | 55.7 | 61.6 |
| | AKS | 65.6 | 68.6 | 60.0 | 52.7 | 58.8 | 65.6 | 70.3 | 60.7 | 55.0 | 60.2 | 65.6 | 73.3 | 60.7 | 54.6 | 60.4 |
| | **SCOPE** | 66.7 | **74.4** | **62.4** | 53.0 | **60.6** | 66.1 | 71.5 | 62.9 | **57.6** | **62.2** | 66.1 | 71.5 | 61.9 | **59.0** | **62.5** |
| **32B** | UNI | **72.0** | **72.7** | 54.4 | 50.9 | 57.7 | **74.6** | 73.8 | 57.5 | 52.1 | 59.8 | **77.2** | 75.6 | 56.1 | 57.4 | 62.2 |
| | TOPK | 68.3 | 66.3 | 57.8 | 54.8 | 59.1 | 68.3 | **75.0** | 59.7 | 57.3 | 61.9 | 68.3 | 75.6 | 61.9 | 58.0 | 62.9 |
| | AKS | 68.3 | 71.5 | **58.5** | 56.4 | 60.7 | 68.3 | 71.5 | 61.7 | 55.7 | 61.3 | 68.3 | 75.6 | **62.1** | 57.1 | 62.6 |
| | **SCOPE** | 69.8 | 72.1 | **58.5** | **58.0** | **61.6** | 68.8 | 73.8 | **62.1** | **60.5** | **63.9** | 68.8 | 75.0 | 61.7 | **60.8** | **64.0** |
| **72B** | UNI | 75.1 | 69.8 | 57.8 | 49.6 | 58.3 | **77.8** | 73.3 | 57.3 | 52.8 | 60.4 | **79.9** | 76.7 | 60.9 | 54.1 | 62.8 |
| | TOPK | 74.1 | 69.2 | 59.7 | 55.9 | 61.3 | 74.1 | 72.7 | **62.9** | 58.5 | 63.9 | 74.1 | 73.3 | 63.1 | 60.3 | 64.8 |
| | AKS | 74.1 | 70.9 | 60.2 | 55.0 | 61.3 | 74.1 | 72.1 | 57.3 | 56.0 | 61.0 | 74.1 | 73.3 | 62.1 | 57.3 | 63.2 |
| | **SCOPE** | **75.7** | **73.3** | **62.4** | **57.4** | **63.6** | 74.1 | 71.5 | **62.9** | **60.1** | **64.4** | 74.1 | 73.3 | **65.3** | **62.8** | **66.5** |

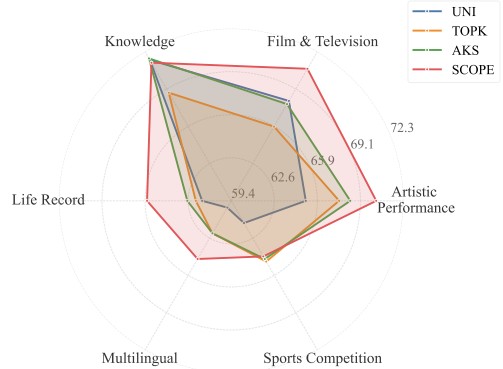

*Figure 3.* Fine-grained performance comparison on Video-MME categories (Qwen2.5-72B, $K = 32$). SCOPE demonstrates superior reasoning capabilities in complex domains such as Film & Television and Knowledge.

**Results on LongVideoBench.** Table 3 details the performance across different duration intervals. SCOPE's advantage becomes increasingly pronounced as video duration grows, effectively **countering "semantic submergence."** In the extreme long-video interval, traditional Top-K methods often falter due to "flat" matching, being distracted by visually prominent but logically irrelevant segments. In contrast, leveraging DAG-structured decomposition, SCOPE maintains robust performance in this challenging interval. Notably, with the 32B model at $K = 64$, SCOPE reaches **60.8%**, far exceeding Top-K, confirming its ability to precisely locate sparse key evidence in massive temporal contexts.

To comprehensively evaluate the robustness and generalization capability of SCOPE, we conducted extensive supplementary experiments. These investigations include ablation studies on different edge-side visual encoders, sensitivity analysis across varying video sampling rates, and generalization tests with different planner and inference backbones. Detailed results and analyses are provided in **Appendix B**. Additionally, we provide qualitative case studies to visualize the step-by-step reasoning process in **Appendix A**.

### 5.2. Inference Efficiency & Pareto Frontier

We verify that our accuracy gains do not come at the cost of prohibitive latency by analyzing the trade-off between edge processing time and reasoning accuracy. Table 4 presents a comprehensive comparison across all baselines using the 72B model.

**Uniform** sampling is a strong temporal-coverage baseline and has negligible selection cost, but its context-agnostic nature prevents it from prioritizing query-critical evidence. Flat semantic methods (**Top-K** and **AKS**) use query-aware similarity scores, but they may concentrate the budget around locally high-similarity segments and miss temporally distributed evidence. **SCOPE** incurs a marginal overhead due to sub-query matrix operations, yet this overhead yields the highest accuracy across all budgets by combining temporal coverage with structured evidence prioritization. Consequently, SCOPE successfully **advances the Pareto frontier**, offering the best accuracy among semantic-aware methods while maintaining comparable latency.

*Table 4.* Edge latency vs. accuracy trade-off on Video-MME using the 72B model. Results are averaged over benchmark samples under the default 1 FPS setting. SCOPE achieves the highest accuracy with only marginal latency overhead compared to semantic baselines.

| Budget | UNI | | TOPK | | AKS | | SCOPE | |
|---|---|---|---|---|---|---|---|---|
| | Acc (%) | Lat (s) | Acc (%) | Lat (s) | Acc (%) | Lat (s) | Acc (%) | Lat (s) |
| 16 | 62.6 | 0.32 | 62.9 | 14.26 | 64.2 | 14.57 | **66.0** | 15.26 |
| 32 | 65.8 | 0.34 | 65.7 | 14.44 | 67.1 | 14.68 | **68.4** | 15.45 |
| 64 | 69.4 | 0.32 | 68.6 | 14.39 | 69.4 | 14.83 | **70.0** | 15.38 |

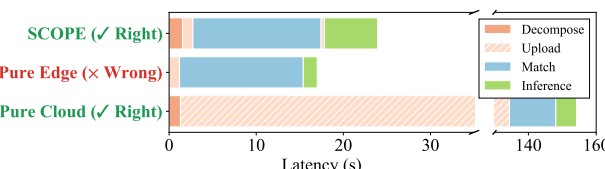

*Figure 4.* **Latency Breakdown & Outcome Comparison.** The Pure Cloud timeline is truncated (broken axis) due to the massive overhead of full-video transmission. SCOPE (bottom) shifts the heavy matching process to the edge, replacing video upload with a lightweight Keyframe Upload phase to achieve the optimal balance.

## 5.3. Deployment Paradigm Comparison

To demonstrate systemic advantages, we compare SCOPE against two baselines on Video-MME: **Pure Cloud**, which executes the full pipeline on the server necessitating complete video transmission, and **Pure Edge**, which processes videos locally using a lightweight 3B model. Quantitative results in Table 5 reveal the critical trade-off: Pure Edge is efficient but yields a low accuracy of 55.29%, whereas Pure Cloud achieves high precision but incurs a prohibitive latency exceeding 150 seconds due to transmission bottlenecks. SCOPE uses the same 72B cloud reasoner as Pure Cloud with deterministic decoding; when the selected keyframes preserve the necessary evidence, the final predictions can coincide with the Pure Cloud setting. Under the 16-frame budget, SCOPE matches the Cloud's 66.04% accuracy while compressing total latency to just **23.94s**—a reduction of approximately **85%**. Figure 4 visualizes the $K = 16$ setting from Table 5. Unlike Pure Cloud, which is bottlenecked by massive video transmission (indicated by the truncated axis), SCOPE shifts semantic matching to the edge. This design replaces raw video upload with a lightweight **Keyframe Upload**, successfully offloading visual perception without compromising reasoning depth.

## 5.4. Ablation Study

To validate the contribution of each core component in SCOPE, we conducted ablation studies using the 3B model backbone. We compared three variants: (1) **w/o Decomp**: Removing the cloud planning module and reverting to standard Top-K retrieval; (2) **w/o Alloc**: Retaining decompo-

*Table 5.* End-to-end performance comparison across deployment paradigms on Video-MME. Results are averaged over benchmark samples under the default 1 FPS setting.

| Budget | Pure Cloud (72B) | | Pure Edge (3B) | | SCOPE (Edge-Cloud) | |
|---|---|---|---|---|---|---|
| | Acc (%) | Lat (s) | Acc (%) | Lat (s) | Acc (%) | Lat (s) |
| 16 | 66.04 | 154.22 | 55.29 | 17.03 | **66.04** | 23.94 |
| 32 | 68.41 | 160.88 | 57.52 | 18.49 | **68.41** | 30.89 |
| 64 | 70.03 | 166.18 | 59.18 | 21.19 | **70.03** | 37.35 |

*Table 6.* Ablation study on query decomposition and budget allocation strategies.

| Budget | w/o Decomp (Top-K) | | w/o Alloc (Random) | | SCOPE (Full) | |
|---|---|---|---|---|---|---|
| | Acc (%) | Lat (s) | Acc (%) | Lat (s) | Acc (%) | Lat (s) |
| 16 | 55.04 | 14.26 | 56.08 | 15.19 | **56.32** | 15.26 |
| 32 | 56.63 | 14.44 | 57.17 | 15.03 | **59.33** | 15.45 |
| 64 | 58.51 | 14.39 | 59.78 | 15.03 | **60.07** | 15.38 |

sition but replacing our budget allocation algorithm with random frame assignment; (3) **SCOPE**: The full framework.

Table 6 reveals that the decomposition mechanism is the primary driver of performance. Even with random allocation, decomposing the query improves accuracy by approximately 1.04% over the flat Top-K baseline at $K = 16$, proving that structured sub-queries retrieve more relevant evidence. Furthermore, incorporating our semantic-driven budget allocation yields an additional gain, confirming that weighted resource distribution is essential for maximizing information gain. Notably, these algorithmic improvements incur negligible latency overhead, ensuring the system remains efficient.

## 6. Related Work

### 6.1. Video-LLMs for Long Video Understanding

Video Large Multimodal Models (Video-LLMs) have rapidly advanced video understanding toward interactive and reasoning-centric settings. Video-LLaMA (Zhang et al., 2023), Video-ChatGPT (Maaz et al., 2024), and Video-LLaVA (Lin et al., 2024) align video representations with large language models to support multimodal question answering and richer temporal reasoning. To scale to long videos, MovieChat (Song et al., 2024; 2025) introduces sparse memory to reduce long-context overhead, while Video-XL (Shu et al., 2025) targets hour-scale understanding. Apollo (Zohar et al., 2025) further analyzes key design choices and bottlenecks in video multimodal modeling.

Despite strong cloud-side performance, the compute, latency, and bandwidth costs of Video-LLMs grow quickly with video length, making direct edge deployment impractical and motivating edge–cloud collaboration.

### 6.2. Edge–Cloud Collaborative Long Video Understanding

Edge–cloud collaboration balances accuracy and efficiency via selective transmission. Approaches include cascaded inference (Ghosh et al., 2023; Nan et al., 2023) and learning-based schedulers using multi-agent RL (Qian et al., 2024; Gao et al., 2024) or accuracy-centric trade-offs (Kong et al., 2023). Addressing long-horizon benchmarks (Fu et al., 2025; Mangalam et al., 2023; Ma et al., 2025a), recent methods employ external memory (Kahatapitiya et al., 2025), retrieval augmentation (Luo et al., 2026; Ma et al., 2025b), or agent-guided frame selection (Wang et al., 2024b; Tang et al., 2025a). However, these approaches remain largely *video-centric*, relying on heuristic scores. While explicit query planning improves controllability in general multimodal reasoning (Yang et al., 2023; Surís et al., 2023), such *query-centric* strategies have not been systematically integrated into edge–cloud video understanding.

## 7. Conclusion

In this paper, we proposed SCOPE, an edge-cloud framework that resolves the bandwidth-accuracy dilemma in long video understanding. By shifting to "query-centric" semantic planning via structured DAGs, SCOPE effectively mitigates "semantic submergence." Real-world experiments confirm that SCOPE advances the Pareto frontier, matching cloud-level accuracy with an approximately 85% latency reduction compared to pure cloud solutions, establishing a new paradigm for semantic-aware distributed inference.

## 8. Limitations

SCOPE has several limitations. First, it is evaluated primarily on verifiable video question answering tasks, where the required visual evidence can often be decomposed into atomic checks. For more open-ended queries, the evidence boundary may be ambiguous, and the planner may generate incomplete or overly broad sub-queries. Second, our current system reduces bandwidth mainly through temporal keyframe selection. Spatial downsampling and video compression are not explicitly optimized in this work, and combining them with query-guided selection remains an important direction for deployment-oriented optimization. Third, the efficiency results depend on the available edge hardware and the quality of the edge-side visual encoder. Although our lightweight encoder uses about 1.4 GB of GPU memory in our implementation, substantially weaker devices or weaker visual encoders may lead to higher edge latency or less accurate frame-query matching.

## Acknowledgments

This work was supported in part by National Natural Science Foundation of China (Grant No. 62522118, 62371313), in part by the Shenzhen Science and Technology Program (Grant No. JCYJ20241202124934046), in part by Guangdong Young Talent Research Project (Grant No. 2023TQ07A708), in part by Shenzhen Loop Area Institute (Contract No. SLAI2026020007).

## Impact Statement

This paper presents work whose goal is to advance the field of machine learning. There are many potential societal consequences of our work, none of which we feel must be specifically highlighted here.

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

## A. Qualitative Case Study

We analyze two representative cases to illustrate how SCOPE's decomposition mechanism guides the retrieval of sparse critical evidence.

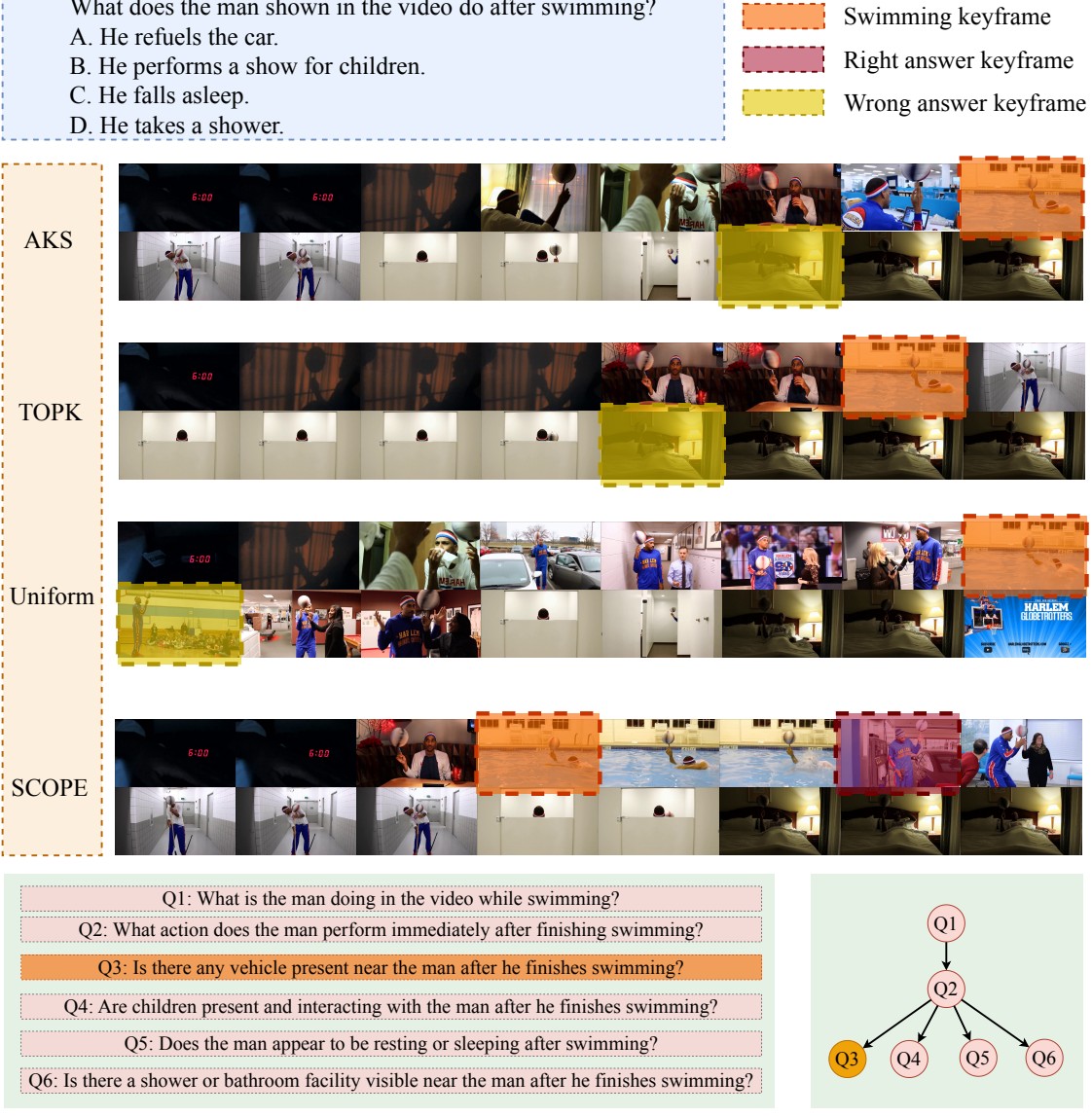

*Figure 5.* **Qualitative Analysis on Causal Reasoning.** The query asks for an action *after* swimming. While baselines focus on redundant indoor scenes, SCOPE generates a specific sub-query regarding "vehicles," enabling it to retrieve the critical frame where the man refuels a car.

**Case 1: Causal Temporal Reasoning.** Figure 5 illustrates a query requiring sequential logic: "*What does the man do after swimming?*". This task demands precise temporal localization of the "swimming" event and the subsequent action. Standard methods like Top-K and AKS are distracted by the visually dominant indoor scenes (e.g., the bedroom), missing the transition point. In contrast, SCOPE decomposes the query into specific sub-goals, including a pivotal check: "*Is there any vehicle present near the man after he finishes swimming?*". Guided by this sub-query, SCOPE allocates a higher budget to the post-swimming segment and successfully retrieves the specific frame where the man refuels a car. This allows the inference model to correctly identify the action "refuels the car," whereas baselines fail to capture this fleeting moment.

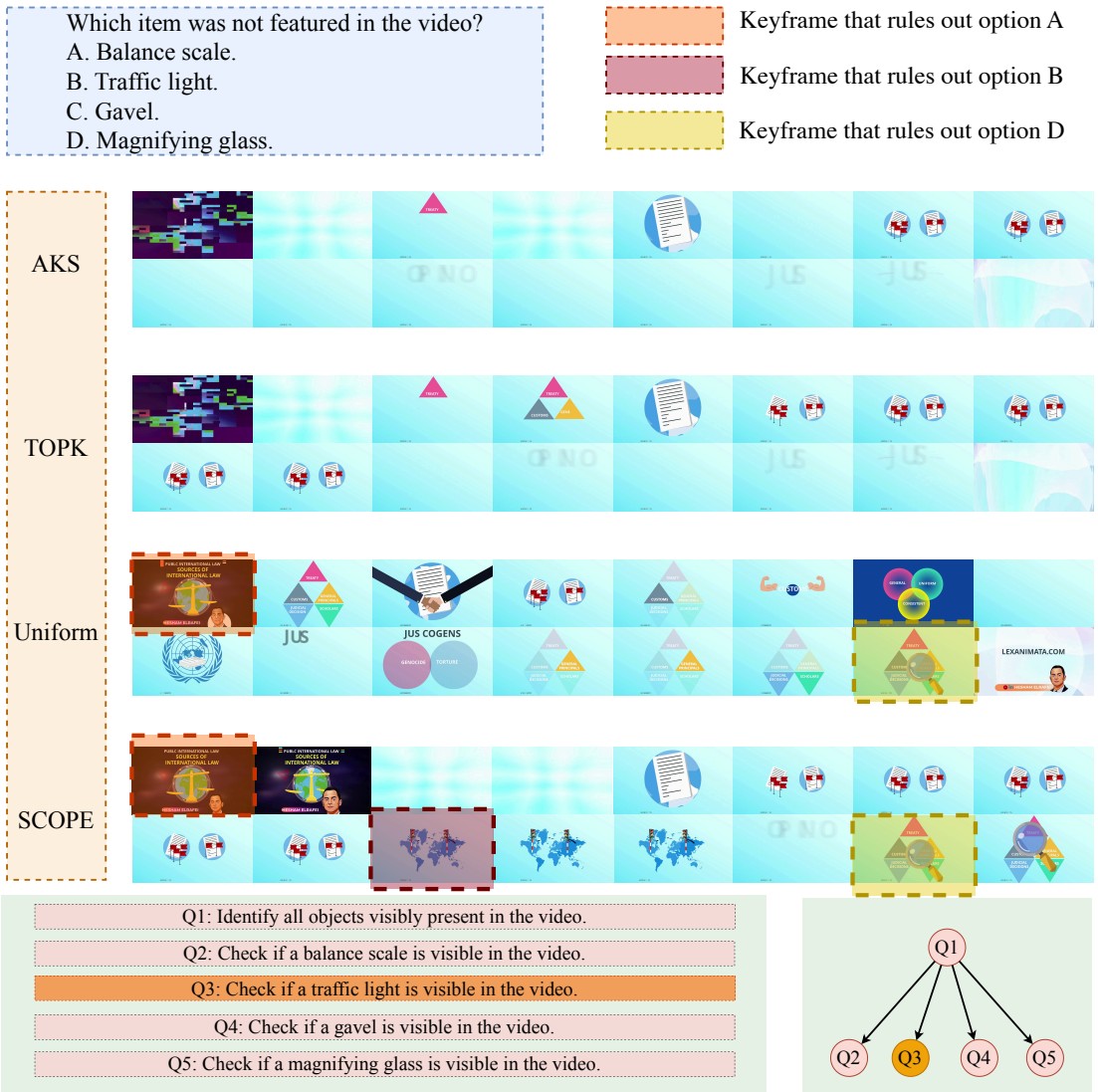

*Figure 6.* **Qualitative Analysis on Exclusionary Reasoning.** The task requires identifying the *missing* item. SCOPE decomposes the query into explicit checks for each option (A, B, D), successfully retrieving the evidence needed to rule them out and correctly selecting Option C (Gavel).

**Case 2: Exclusionary Reasoning.** Figure 6 presents a challenging "negative existence" task: "*Which item was not featured in the video?*". Answering this requires a process of elimination—finding evidence for the presence of distractors to rule them out. Baseline methods (AKS and Top-K) struggle to retrieve frames containing the specific objects (Balance scale, Traffic light, Magnifying glass) because the global query lacks fine-grained object semantics. Interestingly, the context-agnostic Uniform sampling coincidentally captures the "Balance scale" (Option A) and "Magnifying glass" (Option D), but misses the "Traffic light" (Option B). SCOPE, however, explicitly decomposes the options into verification sub-queries (e.g., "*Check if a traffic light is visible*"). This directed search successfully retrieves evidence for Options A, B, and D, enabling the model to confidently identify "Gavel" (Option C) as the correct unfeatured item through logical elimination.

## B. Supplementary Experiments & Analysis

In this section, we conduct extensive ablation studies to verify the component-level contribution, robustness, and generalization capability of SCOPE. Unless otherwise specified, experiments are conducted on the Video-MME dataset with a frame budget of $K = 16$.

## B.1. Impact of Visual Encoders

The edge-side visual encoder determines the quality of semantic matching. We compared the discriminative **CLIP-ViT-B/16** (Default) against the generative **BLIP (Base)** across all inference model scales on both Video-MME and LongVideoBench.

As detailed in Table 7, the choice of visual encoder yields distinct performance trends across benchmarks due to their underlying pre-training paradigms. **BLIP**, trained on cleaner, object-centric data, excels in capturing fine-grained details. Consequently, it consistently outperforms CLIP on **LongVideoBench**, where queries often target specific object states or actions. In contrast, **CLIP**, pre-trained on massive generic image-text pairs, demonstrates superior capability in capturing global semantics and scene-level context. This aligns better with the diverse nature of **Video-MME**, which often involves global perception or multi-scene integration. As a result, CLIP achieves comparable or even superior performance on Video-MME, while maintaining lower computational complexity than generative encoders. A substantially weaker encoder would naturally reduce frame-query matching quality; nevertheless, SCOPE's decomposition and minimum per-sub-query budget provide a simple safeguard against complete starvation of individual evidence types.

*Table 7.* Performance comparison of Visual Encoders (Uniform vs. CLIP vs. BLIP) on Video-MME and LongVideoBench ($K = 16$ for efficiency tests). BLIP shows advantages in object-centric tasks (LongVideoBench), while CLIP remains robust for global context (Video-MME).

| Model | Video-MME Acc (%) | | | LongVideoBench Acc (%) | | |
|---|---|---|---|---|---|---|
| | Uniform | CLIP | BLIP | Uniform | CLIP | BLIP |
| 3B | 54.5 | 56.3 | **57.6** | 52.3 | 57.3 | **58.4** |
| 7B | 56.1 | **60.4** | 60.2 | 56.2 | 60.6 | **62.8** |
| 32B | 60.0 | 63.0 | **63.1** | 57.7 | 61.6 | **63.4** |
| 72B | 62.6 | **66.0** | 65.4 | 58.3 | 63.6 | **65.3** |

## B.2. Sensitivity to Video Sampling Rate

We analyzed the system's robustness under varying video sampling rates ($FPS \in \{0.1, 0.5, 1.0, 2.0\}$) using the 7B model.

As shown in Table 8, accuracy peaks at **1.0 FPS** (60.40%). Interestingly, increasing the sampling rate to 2.0 FPS leads to a slight performance degradation (59.48%), suggesting that an excessive candidate pool may introduce noise that interferes with semantic ranking. Conversely, lowering the rate to 0.5 FPS maintains competitive accuracy (59.96%). However, further dropping to **0.1 FPS** results in a noticeable accuracy decline to 55.96%, indicating that such extreme sparsity begins to compromise the capture of critical visual evidence, despite offering the lowest latency (7.52s). Thus, 1.0 FPS represents the optimal sweet spot, balancing information capture with computational efficiency.

*Table 8.* Impact of sampling rates (FPS) on accuracy and edge latency (Model: 7B). Note that 1.0 FPS achieves the best balance, while higher sampling rates (2.0 FPS) introduce noise that slightly degrades performance.

| Sampling Rate | Acc (%) | Edge Lat (s) |
|---|---|---|
| 0.1 FPS | 55.96 | 7.52 |
| 0.5 FPS | 59.96 | 11.23 |
| **1.0 FPS (Default)** | **60.40** | **15.26** |
| 2.0 FPS | 59.48 | 25.23 |

## B.3. Architectural Generalization

A key advantage of SCOPE is its *model-agnostic* nature. To verify this, we replaced the core LLM components with state-of-the-art alternatives (DeepSeek-V3 and Kimi K2.5 for planning, and InternVL2 for inference).

**Planner Generalization.** We replaced the default Qwen3-80B planner with **DeepSeek-V3** and **Kimi K2.5**. As shown in Table 9, both planners achieve competitive performance, confirming that SCOPE is not coupled to a specific planner and functions effectively with other top-tier reasoning models.

**Inference Generalization.** We replaced the inference engine with **InternVL2-8B**. Notably, the system achieves an impressive accuracy of **63.88%**, surpassing the default Qwen2.5-VL-7B backend. This demonstrates that the "golden

frames" identified by our planning mechanism are universally valuable and can unlock the potential of various multimodal architectures.

*Table 9.* Generalization across different Planner and Inference backbones on Video-MME ($K = 16$). SCOPE demonstrates robust compatibility across different model combinations.

| Cloud Planner | Inference Model | Acc (%) |
|---|---|---|
| **Qwen3-80B (Default)** | Qwen2.5-VL-7B | 60.40 |
| DeepSeek-V3 | Qwen2.5-VL-7B | 59.92 |
| Kimi K2.5 | Qwen2.5-VL-7B | 60.80 |
| Qwen3-80B | **InternVL2-8B** | **63.88** |

## B.4. Planning Overhead and Query Complexity

The cloud-side planner operates on text-only inputs and therefore contributes a small fraction of the total video reasoning latency. Table 10 reports the average decomposition time and output length under the default planner. To further characterize how planning complexity scales with query difficulty, we use a 1–10 LLM-judge difficulty score and report the corresponding average decomposition length, DAG node count, and DAG depth on Video-MME in Table 11. The decomposition complexity increases from simple to difficult queries and saturates for high-difficulty cases, consistent with the planner producing a compact set of visually verifiable sub-goals.

*Table 10.* Average cloud planning overhead and decomposition length.

| Benchmark | Time (s) | Tokens |
|---|---|---|
| Video-MME | 1.3 | 240.41 |
| LongVideoBench | 1.5 | 287.76 |

*Table 11.* Query difficulty vs. decomposition complexity on Video-MME.

| Difficulty | Tokens | DAG Nodes | DAG Depth |
|---|---|---|---|
| 1 | 131.6 | 2.00 | 1.50 |
| 2 | 183.7 | 2.78 | 1.83 |
| 3 | 244.3 | 3.70 | 2.02 |
| 4 | 241.2 | 3.70 | 1.94 |
| 5 | 230.1 | 3.52 | 2.09 |
| 6 | 270.5 | 4.17 | 2.09 |
| 7 | 281.0 | 4.28 | 2.19 |
| 8 | 275.7 | 4.20 | 2.32 |
| 9 | 268.5 | 4.56 | 2.62 |
| 10 | 274.6 | 4.68 | 2.71 |

## B.5. Sensitivity to Importance Scores

SCOPE uses discrete 1–10 importance scores because the resulting allocation is stable and interpretable. We test this design by perturbing the planner-generated scores with Gaussian noise and by changing the scoring range from 1–10 to 1–100. As shown in Table 12, the final accuracy remains stable under both variants, indicating that the decomposition structure and minimum-budget guarantee are more important than small numerical differences in the score scale.

## B.6. Additional Generalization Results

We further evaluate whether the same query-centric selection mechanism transfers to newer inference backbones. Table 13 reports results with Qwen3-VL on Video-MME and LongVideoBench. The numbers are consistent across model scales and datasets: SCOPE achieves the best accuracy in all settings, and the gains are particularly clear on LongVideoBench with the stronger Qwen3-VL-32B backend.

*Table 12.* Sensitivity to importance-score perturbations on Video-MME.

| Budget | Original | Gaussian Noise | 1–100 Scoring |
|--------|----------|----------------|---------------|
| 16 | 66.0 | 65.9 | 65.9 |
| 64 | 70.0 | 69.5 | 69.8 |

*Table 13.* Additional generalization results on newer backbones and benchmarks. All entries report accuracy (%).

| Setting | Budget | Uniform | Top-K | AKS | SCOPE |
|---------|--------|---------|-------|-----|-------|
| Qwen3-VL-8B / Video-MME | 16 | 61.0 | 60.5 | 61.6 | **63.0** |
| Qwen3-VL-8B / Video-MME | 32 | 63.4 | 63.2 | 64.6 | **65.1** |
| Qwen3-VL-8B / Video-MME | 64 | 65.8 | 66.0 | 67.4 | **67.5** |
| Qwen3-VL-8B / LongVideoBench | 16 | 56.2 | 59.2 | 58.5 | **60.7** |
| Qwen3-VL-8B / LongVideoBench | 32 | 58.6 | 60.7 | 60.2 | **63.2** |
| Qwen3-VL-32B / LongVideoBench | 16 | 59.6 | 61.7 | 61.5 | **65.4** |
| Qwen3-VL-32B / LongVideoBench | 32 | 61.2 | 63.5 | 62.5 | **66.8** |

# C. System Prompts

In this section, we provide the full system prompt used for the Query Decomposition module in the Cloud Planner.

---

**System Prompt: Semantic Query Decomposition**

---

**Role:** You are a video question decomposition assistant. Your task is to decompose a complex user query into atomic sub-queries (nodes) and establish temporal dependencies (edges) between them.

---

**Core Requirements:**
- **Decomposition:** Break the question into 3∼6 semantically atomic sub-queries.
- **Importance Scoring:** Assign a score (1-10) to each sub-query:
    – *1-3: Optional*, *4-6: Moderate*, *7-8: Important*, *9-10: Critical*.
- **Output Format:** Strictly output valid JSON only. No markdown or explanations.

**JSON Schema Template:**
```
{
    "nodes":  [
      { "id":  "Q1", "text":  "<description>", "importance":  8, "layer":  0 },
      { "id":  "Q2", "text":  "...", "importance":  6, "depends_on":  ["Q1"] }
    ],
    "edges":  [
      { "parent":  "Q1", "child":  "Q2", "temporal":  "AFTER", "window_s":  0 }
    ]
}
```

**Field Constraints:**
- `nodes[].layer`: Integer, root starts at 0. Same layer implies parallel execution.
- `edges[].temporal`: One of `"AFTER"`, `"BEFORE"`, `"AROUND"`.
    – If `"AROUND"`, `window_s` must be $\geq 20$.
- `edges[].scope`: `"all"` (intersection) or `"any"` (union) for multiple parents.

---

**Input:** `"""{question}"""`

---

