# OpenReview forum: "Think in Cloud, Look at Edges: Semantic-Driven Query Decomposition for Efficient Video Reasoning"
_ICML.cc/2026/Conference — ICML 2026 spotlight_

### Official Review · Reviewer_yB2A · 2026-02-27

**Soundness:** 3
**Presentation:** 4
**Significance:** 3
**Originality:** 3
**Overall Recommendation:** 5
**Confidence:** 5

**Summary:**

This paper proposes SCOPE, a semantic-driven edge–cloud collaborative framework for long video reasoning. The key idea is to shift reasoning to the cloud: a large multimodal model decomposes a complex query into a structured Direct acyclic graph of atomic sub-queries with associated importance scores, all done via prompting engineering. These weighted sub-queries guide the edge device in selecting keyframes under a transmission budget using a Largest Remainder-based allocation strategy. The selected frames are then sent back to the cloud for final reasoning.

**Compliance With Llm Reviewing Policy:**

Affirmed.

**Final Justification:**

The rebuttal thoroughly addresses my concerns with clear and concrete improvements. In particular, the additional experiments on newer backbones (e.g., Qwen3-VL) and datasets (e.g., LVBench) strengthen confidence in the generalizability of the approach, while the planned revisions to clarify experimental details and moderate claims improve the paper’s clarity and rigor. While the explanation of Uniform vs. Top-K is not convincing, my concerns have been satisfactorily resolved, and I have accordingly increased my score.

**Key Questions For Authors:**

1) Why is uniform sampling almost always superior to top-K? This looks a bit counter-intuitive. Do you have any justification for this?
2) Table 5) and Table 6) are based on which data? Could you clarify? Is it based on a single data sample or is it an average over multiple samples?

**Limitations:**

yes

**Strengths And Weaknesses:**

### Strengths

* The paper is well motivated, sound, clear and well written.
* The approach is clear and well formulated.
* Algorithmic clarity.
* Results on VideoMME reveal improved performance in SCOPE methodology.
* The paper presents and cloud-edge hybrid framework with a good trade-off between accuracy and latency empirically validated over pure edge and pure cloud methods.
* The paper presents ablation studies and qualitative results which is always a must have.


---

### Weaknesses

* Authors present results with Qwen2.5VL, however, the Qwen3VL is already available. This is not a main criticism but more a suggestion to include in the main paper.
* The method could benefit from more baselines and extremely long datasets such as LVBench [1] or MF2 [2] to better access its capabilities.
* In line 378, the manuscript argues that Uniform sampling suffers from suboptimal accuracy due to its context-agnostic nature. However, the reported improvements of SCOPE over Uniform sampling are comparatively small. It may be more accurate to temper this statement or quantify the practical significance of the observed gains.
* Section 5.3 misses experimental details. For instance which data you are using, etc.
* I've found the related work a bit outdated and incomplete. While this is not a critical criticism it would benefit from a thoughtful reconsideration.


[1] LVBench: An Extreme Long Video Understanding Benchmark, Wang et al. 2024.

[2] Movie Facts and Fibs (MF2): A Benchmark for Long Movie Understanding, Zaranis et al. 2025

---

> ### Author Rebuttal · Authors · 2026-03-31
>
> We sincerely thank the reviewer for the positive and careful assessment of our paper. We especially appreciate the recognition that the paper is well motivated, technically sound, clearly formulated, and that SCOPE achieves a strong accuracy-latency trade-off in edge-cloud long-video reasoning. We also thank the reviewer for the constructive suggestions regarding newer backbones, broader benchmark coverage, the interpretation of Uniform vs. Top-K, and the missing experimental details around Tables 5-6. Below, we provide a point-by-point response.
>
> # [W1&W2&W5] Newer backbones, broader benchmarks, and related work
> Thank you for your insightful comment. Due to the limited rebuttal time and computing resources, we conducted preliminary additional experiments to further test the extensibility of SCOPE.
>
> **For W1**, we agree that evaluating a newer inference backbone is meaningful. We therefore ran preliminary experiments with Qwen3-VL-8B under frame budgets of 16 and 32 on Video-MME. The results show that SCOPE still consistently outperforms the baselines, suggesting that the benefit of structured query decomposition is not tied to Qwen2.5-VL alone and can transfer to newer VLM backbones.
>
> |Frame|16|32|
> |-|-|-|
> |SCOPE|62.96|65.07|
> |AKS|61.62|64.59|
> |Uniform|61.00|63.41|
> |TopK|60.51|63.18|
>
> **For W2**, we also conducted preliminary experiments on LVBench using Qwen2.5-VL-3B under a frame budget of 16. Again, SCOPE performs the best and shows a clear advantage over the baseline methods.
>
> |Method|Accuracy|
> |-|-|
> |SCOPE|28.8|
> |AKS|27.0|
> |Top-K|25.6|
> |Uniform|23.6|
>
> These additional results further support the transferability and generalization potential of SCOPE across newer backbones and more challenging datasets. At the same time, due to the limited rebuttal time and available resources, we will include more complete experimental results and analysis in the camera-ready version.
>
> **For W5**, we agree that the related work section can be strengthened. We will refresh it to better cover recent long-video benchmarks and collaborative long-video reasoning methods, including the works pointed out by the reviewer such as LVBench and MF2, as well as newer relevant work in long-video understanding and edge-cloud video reasoning.
>
> # [W3] Improvement over Uniform and wording of the claim
> Thank you for your helpful comment. We will revise the statement to a more tempered version:
> “Uniform is a strong temporal-coverage baseline, but it is context-agnostic and can be surpassed by structured query-aware retrieval, especially on harder long-video reasoning settings.”
>
> # [W4&Q2] Missing experimental details in Section 5.3 and clarification of Tables 5-6
> Thank you for your helpful comment. Tables 5 and 6 are both based on Video-MME, and Figure 4 is a visualization of the K=16 setting from Table 5. Tables 5 and 6 report averages over multiple samples rather than results from a single example. We agree that this should have been stated explicitly near the tables. While the benchmark/model setup and default 1 FPS sampling are already specified in Section 4, the local description around Tables 5-6 is indeed underspecified, and we will revise the captions and surrounding text in the camera-ready version to make this explicit.
>
> # [Q1] Why can Uniform sometimes outperform Top-K
> Thank you for your thoughtful question. In long-video QA, critical evidence is often distributed across multiple time segments, so temporal coverage is essential. Flat Top-K retrieval tends to concentrate the budget on a few locally high-similarity segments and can be overly biased toward dominant concepts, whereas Uniform, although simple, provides more even temporal coverage. This is also reflected in Appendix A, Case 2, where Uniform happens to cover part of the key evidence while flat Top-K misses it. This observation further supports our main motivation: semantic matching alone is insufficient, and the value of SCOPE lies in preserving coverage while introducing structured evidence prioritization.

---

> > ### Author Rebuttal · Reviewer_yB2A · 2026-04-01
> >
> > The rebuttal thoroughly addresses my concerns with clear and concrete improvements. In particular, the additional experiments on newer backbones (e.g., Qwen3-VL) and datasets (e.g., LVBench) strengthen confidence in the generalizability of the approach, while the planned revisions to clarify experimental details and moderate claims improve the paper’s clarity and rigor. While the explanation of Uniform vs. Top-K is not convincing, my concerns have been satisfactorily resolved, and I have accordingly increased my score.

---

> > > ### Author Response · Authors · 2026-04-08
> > >
> > > Thank you for your prompt and thoughtful engagement. We greatly appreciate your detailed feedback and your encouraging reassessment of our work. In particular, your suggestions on clarifying experimental details and further improving the presentation will help strengthen the paper's clarity and rigor, and we will carefully incorporate them into the camera-ready version.

---

### Official Review · Reviewer_vf61 · 2026-03-04

**Soundness:** 3
**Presentation:** 3
**Significance:** 3
**Originality:** 3
**Overall Recommendation:** 4
**Confidence:** 2

**Summary:**

Long video understanding faces a trade-off: cloud-based large multimodal models provide strong reasoning but incur high bandwidth and latency costs, while edge-based systems are efficient yet less accurate. Existing collaborative methods rely on similarity-based filtering, reducing complex queries to flat semantic vectors. This can cause dominant visual features to overshadow subtle but logically important cues. This paper proposes SCOPE (Semantic Cloud-Orchestrated Perception at Edge) addresses this by shifting to a “Think in Cloud, Look at Edges” paradigm. A cloud model decomposes complex queries into a structured Directed Acyclic Graph (DAG), forming an observation plan that guides the edge to retrieve evidence based on logical requirements rather than surface similarity.

**Compliance With Llm Reviewing Policy:**

Affirmed.

**Final Justification:**

This is a solid submission that clearly advances the accuracy–latency frontier on this meaningful problem. However, as the performance is already near the practical ceiling. I therefore maintain my initial assessment of weak accept.

**Key Questions For Authors:**

In Table 5, why do SCOPE and Pure Cloud achieve exactly the same accuracy, even without randomness?

**Limitations:**

The DAG-based observation plan may work well for structured queries, but it is unclear how well it generalizes to more open-ended questions where the required evidence is harder to decompose.

**Strengths And Weaknesses:**

Strengths: The method is reasonable. The idea of decomposing complex queries into a structured observation plan is interesting and brings a certain level of novelty compared to conventional similarity-based approaches.

Weaknesses:
1. The gap between existing methods such as AKS and the cloud-only large model is already small, so the improvement brought by SCOPE over these approaches is also limited.
2. The motivation states that previous work is mainly video-centric, but existing methods already select frames for upload based on the similarity between the query and the video. It would be helpful to more clearly explain what fundamentally distinguishes the proposed “Think in Cloud, Look at Edges” idea from similarity-based approaches such as AKS.

---

> ### Author Rebuttal · Authors · 2026-03-31
>
> We sincerely thank the reviewer for the clear summary of our paper and for recognizing the technical soundness of our approach, as well as the novelty of decomposing complex queries into a structured observation plan. We especially appreciate the reviewer’s request for a clearer distinction between SCOPE and similarity-based collaborative methods such as AKS, as this helps us better clarify our central contribution. Below, we provide a point-by-point response to the reviewer’s concerns.
>
> # [W1] Limited empirical gap over strong similarity-based baselines
> Thank you for your insightful comment. Compared with existing strong baselines, the absolute accuracy gain of SCOPE is indeed not an order-of-magnitude leap. However, we would like to emphasize that the main contribution of this paper is not merely an isolated accuracy gain, but rather the advancement of a better accuracy-latency frontier in practical edge-cloud deployment. According to our additional measurements, compared with algorithms such as AKS, SCOPE introduces only about 1.5 seconds of extra latency on the Video-MME dataset, while yielding a 1.8% accuracy improvement with the 72B model and a 2.5% improvement with the 32B model. The core reason is that SCOPE performs semantic planning from the query side, transforming complex questions into a structured observation plan, and further achieves structured evidence prioritization through importance-aware budget allocation. As shown in Table 4, SCOPE consistently lies on a more favorable frontier among semantic-aware baselines. We will revise the wording in the paper to clarify this point.
>
> # [W2] Fundamental distinction between “Think in Cloud, Look at Edges” and similarity-based methods such as AKS
> Thank you for your thoughtful question. Query-video similarity-based keyframe selection is one of the mainstream ideas in current long-video understanding. However, in terms of methodological essence, methods such as AKS/Top-K still perform frame scoring based on a single holistic query representation, and their optimization focus on: how to select several frames from the video that are most relevant to that flat query. In contrast, the core distinction of SCOPE is that we do not directly perform holistic matching on the original query. Instead, we first conduct semantic planning from the query side, decomposing a complex question into multiple atomic evidence requirements, explicitly modeling their temporal/causal dependencies with a DAG, and further applying structured evidence prioritization to different sub-goals through importance-aware allocation. As a result, what is executed on the edge side is no longer holistic matching for a single query, but evidence retrieval oriented toward multiple sub-queries. We believe this is precisely the fundamental distinction between “Think in Cloud, Look at Edges” and similarity-based selection. Table 1 in the paper shows that, under the same frame budget, structured selection brings consistent gains over flat selection; in addition, Appendix A provides qualitative cases to further illustrate this difference. We will further strengthen this point in the camera-ready version.
>
> # [Q1] Why SCOPE and Pure Cloud achieve exactly the same accuracy in Table 5
> Thank you for your helpful question. The purpose of Table 5 is to distinguish deployment paradigms rather than reasoning backbones. Since Pure Cloud and SCOPE use the same 72B cloud-side reasoner under deterministic decoding, once the keyframes selected by SCOPE preserve the evidence required by the cloud model, the two settings can produce  identical predictions, and thus identical accuracy.
>
> # [L1] Generalization of the DAG-based observation plan to more open-ended questions
> Thank you for your insightful comment. For open-ended questions, since their evidence boundaries are less clear and the decomposition space is more open, query decomposition itself becomes more challenging. At the same time, we believe that the query-centric planning of SCOPE still has potential in such scenarios, because it emphasizes organizing retrieval based on evidence requirements rather than flat similarity; however, this paper mainly focuses on verifiable complex video question answering settings and has not yet conducted dedicated experiments on more open-ended scenarios. We will add this discussion in the camera-ready version and treat it as an important direction for future work.

---

> > ### Author Rebuttal · Reviewer_vf61 · 2026-04-02
> >
> > Thank you for the additional details. The improved accuracy–latency frontier is well motivated; however, since existing methods such as AKS already closely approach cloud-only performance, the incremental gains provided by SCOPE appear limited in terms of practical impact. I will keep my original weak accept score.

---

> > > ### Author Response · Authors · 2026-04-08
> > >
> > > We sincerely thank the reviewer for continuing the discussion and for acknowledging that our accuracy-latency perspective is well motivated. In the spirit of constructive discussion, we would like to cite two objective references from recent literature for additional context. We believe these observations help reflect the current difficulty of further improving these benchmarks and may offer a more complete perspective on the practical value of our work.
> > >
> > > 1. **On the overall remaining headroom of current benchmarks:** As multimodal large models continue to advance rapidly, the remaining headroom on benchmarks such as Video-MME is indeed becoming increasingly narrow. For example, in the recent strong AKS work, under its LLaVA-Video-7B setting, the gain over Uniform sampling on Video-MME is about **0.9%** [1]. Against such a challenging and increasingly saturated backdrop, SCOPE can still achieve an additional **1.8%–2.5%** improvement in our tested settings. This is encouraging to us and suggests that DAG-based edge-cloud planning can still have practical value in realistic deployment scenarios.
> > > 2. **On the potential under strict frame budgets:** In the official Qwen2.5-VL technical report, the maximum number of analyzed frames for the evaluated video benchmarks is capped at 768 [2]. By contrast, SCOPE uses at most 64 frames in our experiments—**less than 10% of that budget**—yet still achieves competitive performance on long-video benchmarks such as LongVideoBench (for reference, 66.5 in our setting, versus 60.7 reported for Qwen2.5-VL-72B in the technical report). The substantially reduced frame budget makes SCOPE a more edge-friendly solution.
> > >
> > > We provide these reference points only to present, as objectively as possible, the effort made by SCOPE under the currently very limited room for further improvement. All of the reviewer’s constructive comments have been extremely helpful for us to reflect on and improve this work. We sincerely thank the reviewer again for the valuable time and thoughtful feedback.
> > >
> > > [1] Tang, X. et al. Adaptive Keyframe Sampling for Long Video Understanding. CVPR, 2025.
> > > [2] Bai, S. et al. Qwen2.5-VL Technical Report. arXiv:2502.13923, 2025.

---

### Official Review · Reviewer_CpUx · 2026-03-07

**Soundness:** 3
**Presentation:** 3
**Significance:** 2
**Originality:** 2
**Overall Recommendation:** 4
**Confidence:** 3

**Summary:**

Using LLM (cloud) to decompose an LLM query into subqueries organized in a graph with importance scores. Then select frames based on the subqueries. The selected frames are sent to the cloud for processing.

**Compliance With Llm Reviewing Policy:**

Affirmed.

**Final Justification:**

Thanks for the clarification, I maintain the score of weak accept, as my primary concerns remain largely unchanged.

**Key Questions For Authors:**

1.	What is the distribution of total candidate frames at the default 1 FPS sampling rate?
2.	Can you report the accuracy when the cloud model processes all sampled frames with no selection or budget constraint? Or would this be too much for the MLLM to handle? This upper-bound result is essential for understanding how much accuracy is sacrificed by any frame selection approach.
3.	What resolution are the keyframes transmitted at between edge and cloud? Can the authors discuss whether downsampling spatially would make sense in reducing transmission cost and processing cost as an alternative to low frame budget?
4.	In Figure 4 and Table 4, the latency comparisons use different subsets of methods. Could you provide a unified end-to-end latency comparison across all methods? Specifically, is the latency saving from Uniform selection sufficient to allow a larger frame budget under the same latency constraint?

**Limitations:**

not discussed. for example, the evaluation assumes RTX 4080 at the edge node, how would it impact the trade-off if the edge node is less powerful compute-wise?

**Strengths And Weaknesses:**

soundness: Experiments are thorough tested on multiple scales of VLM and multiple frame budgets. However, key details like video resolution and total candidate frame counts are missing.

Presentation: Fig 4 & table 4 could be improved if it shows the total latency of all the methods (UNI,TOPK,AKS, Pure Edge, Pure Cloud) in the same graph. Specifically, is the latency saving from uniform selection enough to increase frame budget under the same latency constraint?

Significance: Accuracy performance of SCOPE under same frame budget is mostly above other approaches. However, the paper only explores frame selection as a means of bandwidth reduction, without considering the straightforward alternative of reducing frame resolution or applying compression. also see comments in limitations.

Originality: The paper claims that existing approaches mainly aim to understand the video and select frames this way instead of focusing on decomposing the query. I’m not familiar enough with recent papers in this area to assess originality.

---

> ### Author Rebuttal · Authors · 2026-03-31
>
> We sincerely thank the reviewer for the careful and constructive feedback. We especially appreciate the recognition that our experiments cover multiple VLM scales and frame budgets, and we value the suggestions on evaluation completeness, upper-bound analysis, unified latency reporting, and alternative bandwidth-reduction strategies. Below we provide a point-by-point response.
> # [W1&Q1] Missing experimental details and candidate-frame distribution under 1 FPS
> Thank you for your helpful comment. Under the default sampling rate of 1 frame per second, for Video-MME, the mean/median/90th percentile are 1021/489/2680 frames, respectively; for LongVideoBench, they are 477/210/1261 frames, respectively. Regarding resolution, the dominant format in both datasets is 1280×720, with a small number of videos at lower resolutions. We will explicitly include these dataset statistics in the camera-ready version.
>
> # [W2&Q4] Unified end-to-end latency comparison across methods
> Thank you for your insightful comment. Table 4 and Figure 4 in the paper were intended to further compare the efficiency of keyframe selection, as well as the deployment-level trade-offs under different system paradigms. We agree that a unified end-to-end view provides a clearer comparison. Under the 16-frame setting, the unified comparison is:
>
> |Method|ACC|Latency(s)|
> |-|-|-|
> |SCOPE-EC|66|23.94|
> |SCOPE-C|66|154.2|
> |AKS-EC|64.2|22.27|
> |AKS-C|64.2|153.23|
> |TopK-EC|62.9|21.96|
> |TopK-C|62.9|152.9|
> |UNI-EC|62.6|8.02|
> |UNI-C|62.6|139.5|
> |AKS-E|55.3|17.37|
> |TopK-E|55|17.06|
> |SCOPE-E|55|17.06|
> |UNI-E|54.5|3.12|
>
> It can be seen that SCOPE is able to reduce latency while maintaining accuracy, thereby advancing the Pareto frontier.
>
> Regarding whether lower latency under the same latency constraint can be traded for a larger frame budget, we agree that this is theoretically possible. However, a larger frame budget does not necessarily mean better overall performance. Uniform or naive frame inclusion often introduces irrelevant or redundant frames, increases computational cost, and may cause the model to deviate from the most valuable evidence, making it difficult for accuracy to improve further [1,2]. Therefore, the key issue is how to select frames that are truly useful for reasoning. We will add this unified comparison and discussion in the camera-ready version.
>
> # [W3&Q3] Why only frame selection, and whether spatial downsampling/compression is a meaningful alternative
> Thank you for your thoughtful question. In our current implementation, the selected keyframes are transmitted at their original resolution. In both datasets, the dominant resolution is 1280×720, with a small number of videos at other resolutions. We agree that spatial downsampling or compression is a meaningful and practical direction for reducing bandwidth and processing cost. In this paper, we focus on temporal frame selection so that the effect of query-guided frame allocation can be studied more clearly. This direction is complementary to SCOPE and can be directly integrated with our framework. We will include it as an important future direction in the camera-ready discussion.
>
> # [Q2] Accuracy upper bound when the cloud model processes all sampled frames without selection or budget constraint
> Thank you for your insightful question. We agree that having an upper-bound reference is useful. However, according to the official Qwen report, due to the limitation of the model’s input context, feeding all frames into the model without any selection or budget control is overly difficult [3]. Although the official report provides an evaluation result, that result is also obtained under a constrained frame budget. Moreover, on LongVideoBench, the official reported value is 60.7, while our method achieves 66.5, so we believe that result cannot serve as an upper-bound reference either.
>
> # [L1] Impact of using a less powerful edge device than the RTX 4080 setting assumed in the paper
> Thank you for your thoughtful comment. We agree that if the edge device is significantly weaker, the trade-off between edge and cloud may change. Our edge server uses an RTX 4080 Laptop GPU, which is itself substantially weaker than a desktop RTX 4080. Based on our measurements, the lightweight edge encoder uses around 1.4 GB of GPU memory, which is already within the capacity of most edge devices. With the continued development of hardware, edge-side computing power is also expected to improve over time. We will explicitly add this discussion in the camera-ready version.
>
> [1] Hu, K. et al. M-LLM Based Video Frame Selection for Efficient Video Understanding. CVPR, 2025.
> [2] Yao, L. et al. Generative Frame Sampler for Long Video Understanding. Findings of ACL, 2025.
> [3] Bai, S. et al. Qwen2.5-VL Technical Report. arXiv:2502.13923, 2025.

---

> > ### Author Rebuttal · Reviewer_CpUx · 2026-03-31
> >
> > Thanks for the response.
> > Q1 & Q2. Clarified.
> > Q3 & Q4. My concerns about alternative reduction methods and an absolute latency comparison in place of # frames comparison remain unchanged. My original score was already a weak accept, and I maintain it given unresolved concerns.

---

> > > ### Author Response · Authors · 2026-04-08
> > >
> > > We sincerely thank the reviewer for the continued engagement and the helpful suggestions. In this final rebuttal response, we aim to address lingering concerns, which also helped us observe an interesting phenomenon related to redundant visual context in long-video VLM reasoning, thereby further enhancing the comprehensiveness of our submission.
> > >
> > > **For the concern about absolute latency comparison**, we agree that this is a deployment-relevant perspective: namely, whether Uniform sampling can leverage its lower selection cost to accommodate more frames and thus outperform SCOPE under a similar latency budget. We conducted additional experiments with Qwen2.5-VL-3B, testing larger Uniform sampling frame budgets with comparable or higher latency (128/192/256), and evaluated their accuracy and latency against SCOPE. The results are as follows:
> > >
> > > |Method|LongVideoBench Acc. ↑|LongVideoBench Latency ↓|Video-MME Acc. ↑|Video-MME Latency ↓|
> > > |-|-|-|-|-|
> > > |SCOPE-32|58.6|20.4|59.3|19.3|
> > > |SCOPE-16|57.3|18.6|56.3|17.9|
> > > |Uniform-128|54.5|17.3|58.4|15.8|
> > > |Uniform-192|55.2|25.9|58.1|21.4|
> > > |Uniform-256|54.7|37.9|57.7|35.2|
> > >
> > > We find that, across both datasets, **Uniform sampling does not show consistent accuracy gains as the frame budget increases, while end-to-end latency increases with the transmitted data volume and model input size**. Meanwhile, on LongVideoBench, SCOPE outperforms Uniform sampling in accuracy.
> > >
> > > We believe this suggests that simply converting the latency budget into more unfiltered frames cannot reliably improve long-video reasoning. Directly transmitting a large number of unfiltered frames to the cloud VLM may introduce redundant or irrelevant visual context, which may, in turn, interfere with the reasoning process. The core value of SCOPE is to ensure that the cloud model receives a critical evidence chain that directly supports reasoning.
> > >
> > > **Regarding the reviewer’s concern about whether spatial downsampling can serve as an alternative to using a low frame budget**, we fully agree that downsampling/compression is a practical way to reduce transmission cost. However, we do not view it as a full replacement in long-video reasoning, but rather as a complementary strategy. The reason is that relying only on downsampling may reduce bandwidth cost, but from the perspective of reasoning accuracy, aggressive spatial compression may discard fine-grained visual details that are important for reasoning, thereby degrading performance [1,2].
> > >
> > > On the other hand, even with reduced spatial resolution, in long-video reasoning it is still typically infeasible to feed the entire video to the model, because the practical bottleneck is not only per-frame size but also the total number of frames that the model can effectively process. In fact, the Qwen2.5-VL technical report itself evaluates videos with the maximum number of analyzed frames capped at 768 [3]. Therefore, keyframe selection remains necessary, and query-guided approaches such as SCOPE still have substantial value.
> > >
> > > We hope that this response helps clarify the remaining concerns about our work. Thank you again for your time and effort in reviewing our manuscript and for your valuable comments.
> > >
> > > [1] Wang, W. et al. Divide, Conquer and Combine: A Training-Free Framework for High-Resolution Image Perception in Multimodal Large Language Models. AAAI, 2025.
> > > [2] Li, Z. et al. Monkey: Image Resolution and Text Label Are Important Things for Large Multi-modal Models. CVPR, 2024.
> > > [3] Bai, S. et al. Qwen2.5-VL Technical Report. arXiv:2502.13923, 2025.

---

### Official Review · Reviewer_Wr6x · 2026-03-13

**Soundness:** 4
**Presentation:** 4
**Significance:** 4
**Originality:** 3
**Overall Recommendation:** 4
**Confidence:** 4

**Summary:**

The paper addresses the challenge of efficient long video understanding by proposing SCOPE, a semantic-driven edge-cloud collaborative framework. It introduces a paradigm shift from video-centric compression to query-centric planning, where a cloud-based LMM decomposes complex queries into a structured directed acyclic graph of atomic sub-queries. The edge node then utilizes this observation plan to execute budget-aware keyframe selection using the largest remainder method and parallel semantic matching. Experiments on Video-MME and Long VideoBench demonstrate that SCOPE matches cloud-level reasoning accuracy while reducing end-to-end latency by approximately 85% compared to pure cloud solutions.

**Compliance With Llm Reviewing Policy:**

Affirmed.

**Key Questions For Authors:**

1. What is the average token count and generation time for the DAG decomposition on the cloud, and how does this scale with the complexity of the user query?
2. How does the framework handle ambiguous queries where atomic sub-queries might overlap semantically, potentially leading to redundant keyframe selection?
3. Have you evaluated the impact of using a continuous or higher-resolution importance score instead of the discrete 1-10 scale?
4. How does SCOPE perform when the edge-side visual encoder is significantly less capable than the one used in your experiments?

**Limitations:**

yes

**Strengths And Weaknesses:**

**Strengths**
1. The paper identifies and formalizes the semantic submergence problem, where dominant visual features mask subtle but logically critical cues in traditional flat matching strategies.
2. The proposed DAG-based query decomposition effectively captures complex temporal and causal dependencies, leading to superior evidence retrieval in narrative-heavy videos.
3. SCOPE demonstrates exceptional efficiency, advancing the pareto frontier by providing cloud-grade accuracy with significantly lower transmission costs and latency.

**Weaknesses**
1. There is a marginal latency overhead introduced by the sub-query matrix operations and DAG generation at the edge/cloud compared to simpler semantic matching baselines.
2. The system's performance is heavily dependent on the quality of the cloud planner, and an incorrect decomposition could lead to a starvation of critical evidence despite the budget allocation.
3. The importance scoring is discrete and heuristic-based, and the paper does not deeply explore how sensitive the final accuracy is to small variations in these scores.

---

> ### Author Rebuttal · Authors · 2026-03-31
>
> We sincerely thank the reviewer for the positive assessment of our work. We especially appreciate the recognition of the semantic submergence problem, the value of our DAG-based query decomposition, and SCOPE’s strong latency-accuracy trade-off. Below we respond point by point.
> # [W1] Marginal latency overhead introduced by DAG generation and sub-query matching
> Thank you for your thoughtful comment. Compared with flat Top-K, SCOPE adds modest overhead, but this cost enables more reliable retrieval of logically critical evidence. Under the 72B / 16-frame setting, our additional measurements are as follows:
>
> |Bench|Decomp.(s)|Match(s)|Extra(s)|Acc gain|
> |-|-|-|-|-|
> |Video-MME|1.3|0.5|1.8|3.1%|
> |LongVideoBench|1.5|0.4|1.9|2.3%|
>
> Inspired by the reviewer’s comment, we will clarify this trade-off in the paper.
> # [W2] Dependence on the cloud planner and risk of missing critical evidence under incorrect decomposition
> Thank you for your insightful comment. We agree that planner quality matters. To test robustness, **we replaced the planner with Kimi K2.5; under the same setting as Appendix B.3, the accuracy is 60.8%.** We believe SCOPE is stable across planners for two reasons: the cloud has sufficient resources to run high-quality models, and our prompt makes the decomposition goal explicit—producing sub-queries for CLIP-style matching—which encourages stable outputs. We will clarify this in the paper.
> # [W3&Q3] Discrete and heuristic importance scoring, and sensitivity to score variations
> Thank you for your insightful comment. We use 1-10 importance scores because this kind of discrete scoring has been widely used in prior work as a common approach for interpretability and stable judgment [1,2]. We also ran extra tests on Video-MME by perturbing LLM-produced scores with Gaussian noise and by changing the prompt from 1–10 to 1–100:
> |frames|Original acc|Gaussian noise|1–100 scoring|
> |-|-|-|-|
> |16|66.0|65.9|65.9|
> |64|70.0|69.5|69.8|
>
> At 16 frames, noise changes allocation little because the budget is already small. At 64 frames, noise has a larger effect, but SCOPE’s decomposition and per-sub-query minimum-budget guarantee keep the overall impact limited. The experiments also show that changing to 1–100 scoring has little effect. More complex scoring schemes remain a meaningful future direction, and we will discuss this in the paper.
>
> # [Q1] Average token count and generation time of DAG decomposition, and how they scale with query complexity
> Thank you for your helpful question. In our further measurements, the average cloud planning time is 1.3s on Video-MME and 1.5s on LongVideoBench, with average decomposition lengths of 240.41 and 287.76 tokens. Since current benchmarks do not provide explicit query-complexity labels, we conducted a simple query-complexity evaluation using an LLM Judge. We asked Qwen3-80B to assign each query a difficulty score from 1 to 10, and then analyzed its correlation with the decomposition output token count, the number of DAG nodes, and DAG depth. Results on Video-MME are as follows:
> |Difficulty|tokens|DAG nodes|DAG depth|
> |-|-|-|-|
> |1|131.6|2|1.5|
> |2|183.7|2.78|1.83|
> |3|244.3|3.7|2.02|
> |4|241.2|3.7|1.94|
> |5|230.1|3.52|2.09|
> |6|270.5|4.17|2.09|
> |7|281.0|4.28|2.19|
> |8|275.7|4.2|2.32|
> |9|268.5|4.56|2.62|
> |10|274.6|4.68|2.71|
>
> Overall, query complexity shows a coarse increasing trend in DAG decomposition complexity. Inspired by the reviewer’s question, we will add a discussion of this point in the paper.
> # [Q2] Handling semantically overlapping atomic sub-queries and potential redundant keyframe selection
> Thank you for your insightful question. SCOPE handles semantically overlapping sub-queries in two ways. On the one hand, the cloud planner generates atomic, visually verifiable sub-queries with explicit DAG dependencies, thereby reducing overlap at the planning stage. On the other hand, the edge side performs MMR-based selection within each sub-query to suppress redundant high-scoring frames. Inspired by the reviewer’s comment, we will clarify this in the paper.
> # [Q4] Performance when the edge-side visual encoder is significantly weaker
> Thank you for your helpful question. If the edge-side visual encoder is much weaker, frame-query matching and overall performance are expected to decline, since current long-video frame selection methods rely on strong vision-language representations [3,4]. However, although CLIP is no longer a very new model, it can still achieve strong performance in text-image matching [5]. We will clarify this in the paper.
>
> [1] Hashemi et al. *LLM-Rubric*. ACL 2024.
> [2] Kim et al. *Prometheus 2*. EMNLP 2024.
> [3] Hu et al. *M-LLM Based Video Frame Selection for Efficient Video Understanding*. CVPR 2025.
> [4] Yao et al. *Generative Frame Sampler for Long Video Understanding*. Findings of ACL 2025.
> [5] Jia et al. *Scaling Up Visual and Vision-Language Representation Learning With Noisy Text Supervision*. ICML 2021.

---

> > ### Author Rebuttal · Reviewer_Wr6x · 2026-04-04
> >
> > My concerns have been adequately addressed and I have decided to maintain my original score.

---

> > > ### Author Response · Authors · 2026-04-08
> > >
> > > We sincerely thank the reviewer for the positive reassessment and for the time and effort devoted to reviewing our work. We are glad that our clarifications addressed your concerns, and we will carefully incorporate them in the camera-ready version.

---

### Decision · Program_Chairs · 2026-04-30

**Decision:**

Accept (spotlight)

**Comment:**

All reviewers are in favor of acceptance, praising the method for its unique approach to query decomposition, its efficiency, the thoroughness of the experiments and the accuracy of the results. The paper should generate interest and could have significant impact on the video reasoning research community.